# *In Vitro* Embryo Culture Impacts Heart Mitochondria in Male Adolescent Sheep

**DOI:** 10.3390/jdb13020017

**Published:** 2025-05-13

**Authors:** Reza Amanollahi, Stacey L. Holman, Ashley S. Meakin, Monalisa Padhee, Kimberley J. Botting-Lawford, Song Zhang, Severence M. MacLaughlin, David O. Kleemann, Simon K. Walker, Jennifer M. Kelly, Skye R. Rudiger, I. Caroline McMillen, Michael D. Wiese, Mitchell C. Lock, Janna L. Morrison

**Affiliations:** 1Early Origins of Adult Health Research Group, Health and Biomedical Innovation, UniSA: Clinical and Health Sciences, University of South Australia, Adelaide, SA 5001, Australia; reza.amanollahi@mymail.unisa.edu.au (R.A.); stacey.holman@unisa.edu.au (S.L.H.); ashley.meakin@unisa.edu.au (A.S.M.); monalisa@bindi-international.org (M.P.); k.botting@ucl.ac.uk (K.J.B.-L.); song.zhang@thekids.org.au (S.Z.); severence.maclaughlin@gmail.com (S.M.M.); icmcmillen@outlook.com (I.C.M.); 2South Australian Research and Development Institute, Turretfield Research Centre, Rosedale, SA 5350, Australia; dave.kleemann@sa.gov.au (D.O.K.); walkersa@adam.com.au (S.K.W.); jenkelly@internode.on.net (J.M.K.); skye.rudiger@sa.gov.au (S.R.R.); 3Centre for Pharmaceutical Innovation, Clinical & Health Sciences University of South Australia, Adelaide, SA 5001, Australia; michael.wiese@unisa.edu.au

**Keywords:** ART, *in vitro* embryo culture, cardiac metabolism, male, heart, mitochondria, adolescent sheep

## Abstract

Assisted reproductive technology (ART)such as *in vitro* embryo culture (IVC), is widely used in human infertility treatments; however, its long-term effects on the cardiac health of offspring remain unclear. This study aimed to determine whether the effects of IVC on cardiac metabolism and associated signaling pathways persist after birth into adolescence. Embryos were either transferred to an intermediate ewe (ET) or cultured *in vitro* in the absence (IVC) or presence of human serum (IVCHS) with methionine supplementation (IVCHS+M) for 6 days after mating. Naturally mated (NM) ewes were used as controls. Protein expression and hormone concentrations in the left ventricle (LV) were analyzed using Western blot and LC-MS/MS analyses, respectively. IVC was associated with sex-specific alterations in cardiac mitochondria, with males exhibiting reduced mitochondrial abundance. Cardiac protein expression of oxidative phosphorylation (OXPHOS) complexes 1 and 4 was reduced by IVC. Additionally, IVC reduced protein expression of PDK-4 and Mn-SOD in the IVCHS+M group, which may impact energy efficiency and defense against oxidative stress. These changes may predispose IVC offspring to cardiac oxidative stress and mitochondrial dysfunction, particularly in males. This study provides insights into the sex-dependent effects of IVC on cardiac health, emphasizing the importance of evaluating long-term cardiovascular risks associated with IVC protocols.

## 1. Introduction

In recent decades, the use of *in vitro* fertilization (IVF) and related procedures involving the handling of eggs or *in vitro* embryo culture (IVC), has increased rapidly in both human and livestock reproduction [1,2]. The first human IVC birth occurred in 1978, and many such offspring have now reached adulthood [3]. However, there is contrasting evidence that offspring conceived through IVC may have an increased risk of heart disease, highlighting that alterations in the periconceptional environment can have significant effects on postnatal health [4,5,6,7]. To closely mimic the in vivo environment, human serum was used in IVC in the 1980s [6,8]. Subsequent ruminant studies revealed various developmental abnormalities, including increased heart weight in fetuses exposed to IVC with serum [9,10,11]. This effect was likely due to reduced methylation, suggesting that supplementing the culture media with methionine may help restore methylation and normalize cardiac growth [9]. However, serum-free media have now become the gold standard for optimal preimplantation embryo culture in all species, but there remains a cohort of individuals exposed to human serum during the periconceptional period that is now reaching early adulthood [12,13].

The developmental origins of health and disease (DOHaD) concept suggests that environmental factors during embryo culture may exert additional risk factors for poor adult health outcomes [14,15]. The association between IVC and adverse cardiometabolic outcomes is not fully understood. Studies in mice show that the duration of embryo culture differentially affects cardiovascular and metabolic health, but only in male offspring [16]. Moreover, a number of preclinical animal studies have shown outcomes such as large offspring syndrome, organ defects, and epigenetic changes in *IGF2R*, through which IVC may impact cardiovascular physiology [9,17]. However, recent studies in both humans and animals have reported no alterations in some cardiovascular outcomes from childhood to adulthood of individuals conceived through IVC [7,18,19]. While this finding is reassuring, continued monitoring and further research are necessary to ensure optimal cardiovascular health in adults conceived through IVC.

Cardiovascular development in humans closely resembles that in sheep, as both have a prepartum surge in cortisol and thyroid hormones that promotes the maturation of cardiomyocytes before birth [20,21,22,23,24]. To initiate glucocorticoid (GC) signaling, cortisol must bind to glucocorticoid receptors (GRs), which then promote or repress the transcriptional regulation of various genes [25,26]. In sheep, multiple GR isoforms have been identified in the placenta and lungs [27,28,29]. More recently, we identified five distinct isoforms in the fetal sheep heart [30]; however, the associated signaling pathways within the heart remain largely unexplored. While the prenatal surge in cortisol is known to increase mitochondrial abundance in skeletal muscle [31], its effects on the heart’s mitochondrial profile and associated pathways remain less well studied. Mitochondria play a vital role in producing ATP through the process of oxidative phosphorylation (OXPHOS), consisting of complexes 1 to 5 of the electron transport chain (ETC) [32,33], which is particularly important for the heart. The heart’s primary fuel source shifts from glucose during fetal life to free fatty acids in adulthood, a process that involves glucose transporters (i.e., GLUT-1, and GLUT-4) and fatty acid uptake molecules such as CD36 and CPTI [34,35]. Cardiac contraction is crucial for delivering blood throughout the body, a process regulated tightly by the interaction between the sarcoendoplasmic reticulum calcium ATPase (SERCA) and phospholamban (PLN) proteins [36].

Elevated plasma cortisol concentrations that occur in the presence of prenatal stress or hormone treatments commonly linked to IVC affect fetal heart development [37,38,39]. Despite advances in ARTs, there remains a gap in our knowledge of whether *in vitro* embryo culture and transfer have lasting impacts on postnatal cardiometabolic profile. Therefore, we hypothesized that IVC in the absence or presence of human serum along with methionine supplementation, an intervention aimed at replenishing the methylation levels, will impact cardiac metabolism in sheep offspring in a sex-specific manner. We particularly focused on cardiac hormone concentrations, the expression of GR isoforms, and molecular markers of cardiac metabolism and contractility.

## 2. Materials and Methods

### 2.1. Animal Ethics

The study was conducted according to the IMVS/University of South Australia (#110/07) and the Primary Industries and Resources South Australia (#12/09) Animal Ethics Committee and complied with the Australian code of practice for the care and use of animals for scientific purposes. All investigators adhered to the ethical principles and reporting standards for animal experiments as outlined by Grundy et al. [40] and followed the 3Rs principles, particularly regarding the reduction in animal use in research [41]. The ARRIVE guidelines were also applied where relevant [42].

### 2.2. Animals and Experimental Design

All animals were fed 100% of nutritional requirements (7.6 MJ/day for the maintenance of a 64 kg non-pregnant ewe) defined by the Agricultural and Food Research Council [43]. In order to increase the ovulation rate, donor ewes received a concentrated diet (10.6 MJ/kg DM and 151 g/kg of crude protein) supplemented with 350 g of peas/day for 14 days prior to ovulation. Synchronization and superovulation techniques were induced in the donor ewes according to standard procedures as previously defined [44,45,46]. Ewes were treated with progestagen pessaries (45 mg flugestone acetate; Intervet, France) for 12 days. Commencing 48 h prior to pessary removal, 9.5 mL of follicle-stimulating hormone (FSH; 190 mg NIH-FSH-P1 standard, Follotropin, Bioniche Inc., Belleville, ON, Canada) was administered as two daily injections over three days with 500 IU of equine chorionic gonadotropin (eCG; Pregnecol, Bioniche Inc., Belleville, ON, Canada) administered at the time as the last FSH injection. Twenty-seven hours after pessary removal, synthetic gonadotropin-releasing hormone (GnRH; 30 μg, Fertagyl, Intervet, France) was administered. Both intermediate and recipient ewes were synchronized using progestagen pessary treatment over 12 days, with 400 IU of eCG (Pregnecol, Bioniche Inc., Canada) being administered at the time of pessary removal. The ovulatory cycles of the recipient ewes were synchronized with the ovulatory cycles of donor animals (±12 h) [4,47].

Eighteen hours after the administration of GnRH treatment, donor ewes were artificially inseminated with fresh semen obtained from a proven fertile ram. Approximately 20 × 10^6^ spermatozoa were deposited into the lumen of each uterine horn using a laparoscope. The same ram was employed for all replicates and breeding of naturally mated (NM) ewes. Oviducts were flushed ~12–17 h after the anticipated median time of ovulation to collect zygotes [44,47]. These zygotes were either transferred immediately to an intermediate recipient (ET) or subjected to *in vitro* culture in a defined synthetic oviductal fluid medium [48]. The culture conditions included absence of human serum (*in vitro* culture, IVC), presence of human serum (IVCHS), or presence of human serum with methionine supplementation (IVCHS+M) until day 6 (day 0 = day of laparoscopic insemination). Ewes in the NM group were naturally mated to give 5 experimental groups [7]. Zygotes randomly chosen for in vivo culture were transferred to intermediate recipients through mid-ventral laparotomy (7 to 22 embryos per recipient; 1 recipient per replicate) [49]. Synthetic oviduct fluid medium was prepared as previously outlined [48,50]. The IVC medium was a combination of SOF enriched with bovine serum albumin (4 mg/mL) and amino acids at concentrations similar to those found in oviduct fluid. The IVCHS medium consisted of a mixture of IVC medium and 20% human serum (HS). Fresh HS was prepared for each replicate. To obtain HS, 10 mL of whole venous blood was collected from a male subject and immediately centrifuged at 2000× *g* for 20 min, allowing plasma to clot and serum to be extracted by compressing the clot. Serum was then heat-inactivated at 56.0 °C for 30 min, filtered, and stored at 4.0 °C. The IVCHS medium was created by mixing 16 mL of IVC medium with 4 mL of HS, resulting in a final concentration of 20% HS. For the IVCHS+M medium, 5000 μM of methionine (Sigma–Aldrich, St. Louis, MA, USA) was added as a supplement, representing the maximum amount soluble in the medium. Zygotes assigned to the embryo culture groups (IVC, IVCHS, and IVCHS+M) underwent three washes in their respective medium. They were then transferred to wells in a 4-well culture dish (Thermo Fisher Scientific, Waltham, MA, USA), each containing 600 μL of the designated medium (IVC, IVCHS, or IVCHS+M), covered with 300 μL of sterile mineral oil (Sigma), and cultured for 6 days. *In vitro* embryo culture was conducted in an environment with 5% CO_2_, 5% O_2_, and 90% N_2_, maintained at 38.5 °C, with groups comprising 12–15 embryos. Recipient ewes were randomly assigned to one of four treatment groups: ET (*n* = 11), IVC (*n* = 20), IVCHS (*n* = 11), and IVCHS+M (*n* = 11) [7]. On day 6, single embryos from these groups were transferred via laparoscopy to the uterine horn ipsilateral to an ovary containing a corpus luteum. The control group comprised naturally mated ewes (NM; *n* = 9) carrying singleton fetuses. The feed allowance per singleton pregnancy was progressively increased by 15% every 10 days from gestational day 115 until the ewes naturally gave birth [7].

### 2.3. Post-Mortem and Tissue Collection

At 6 months of age, lambs were humanely killed with an overdose of sodium pentobarbitone (150 mg/kg, Virbac Pty Ltd., Peakhurst, NSW, Australia). The hearts were dissected and weighed, and samples were snap frozen in liquid nitrogen, stored at −80 °C for subsequent molecular studies, or fixed in 4% paraformaldehyde for histological analysis.

### 2.4. Quantification of Cardiac mRNA Expression

RNA extraction from LV tissue (NM = 5 male (M), 4 female (F); ET = 5M, 5F; IVC = 5M, 6F; IVCHS = 4M, 5F; IVCHS+M = 5M, 4F) was performed using QIAzol reagent (Qiagen, Hilden, Germany) with a tissue homogenizer (TissueLyser LT, Qiagen, Hilden, Germany), followed by purification using the RNeasy Mini Kit (Qiagen, Zurich, Switzerland). The quality and concentration of RNA were evaluated by measuring absorbance at 260 and 280 nm, and RNA integrity was verified using agarose gel electrophoresis. cDNA was then synthesized from 1 μg of RNA via reverse transcription using Superscript III and random hexamers (Invitrogen Australia Pty Ltd., Mt Waverley, VIC, Australia). Negative controls without RNA or Superscript III were included to detect potential DNA or reagent contamination [51]. All essential details regarding our procedure have been included, following MIQE guidelines [52]. Quantitative real-time reverse transcription-PCR was employed to assess the expression of target genes relative to three housekeeping genes: *HPRT* [53], *PGK1* [54], and *GAPDH* [54]. This analysis was conducted on a ViiA7 Fast Real-time PCR system (Applied Biosystems, Waltham, MA, USA). The selection of housekeeping genes was based on their stability across treatment groups, as determined using the GeNorm algorithm [53,55]. The primer sets for target genes included the following markers of cardiac metabolism: *GLUT-4*, *GSK3-β*, *PC*, *ACC*, *CPTI*, *CD36*, *PPARα*, *PPARγ*, *PDK-4*, and *PGC-1α* [56,57,58]. In each qRT-PCR reaction well, there were 3 μL of Fast SYBR Green Master Mix (Applied Biosystems), 0.8 μL of H_2_O, 0.6 μL each of forward and reverse primer (GeneWorks, SA, Australia) for the target genes, and 1 μL of diluted relevant cDNA. Each cDNA sample was run in triplicate for every gene, with no-cDNA controls included in each run to check for reagent contamination. Data analysis was performed using DataAssist Software v3.0 (Applied Biosystems, Waltham, MA, USA), and results were reported as mean normalized expression (MNE) [59,60].

### 2.5. Quantification of Cardiac Protein Expression

Approximately 100 mg of LV tissue (NM = 5M, 4F; ET = 5M, 5F; IVC = 5M, 6F; IVCHS = 4M, 5F; IVCHS+M = 5M, 4F) was sonicated (John Morris Scientific, Chatswood, NSW, Australia) in lysis buffer containing tris HCl (50 mM), NaCl (150 Mm), NP-40 (1%), sodium orthovanadate (1 mM), sodium fluoride (30 mM), sodium pyrophosphate (10 mM), EDTA (10 mM), and protease inhibitor (1 tablet/20 mL buffer; Complete Mini, Roche). The suspensions were then centrifuged (Eppendorf Centrifuge 5415, Crown Scientific, NSW, Australia) for 14 min at 14,300× *g* and 4 °C. Protein content of the extracts was quantified using a Micro BCA Protein Assay Kit (PIERCE, Thermo Fisher Scientific, Waltham, MA, USA), with bovine serum albumin (2 mg/mL stock solution) used to generate the standard curve. Extracted protein (5 mg/mL) was resolved using SDS-PAGE gels, which were stained with Coomassie blue. No significant differences in the abundance of major proteins were observed between samples [53,57,61]. Due to the large number of samples and to have the power to test for an effect of sex, males and females were randomly allocated across two gels, and then each animal was normalized to an internal control (pooled sample) for each gel. The separated proteins were then transferred onto a nitrocellulose membrane (0.45 µm; Hybond ECL, GE Healthcare, Mascot, NSW, Australia). To check the quality of the transfer and stain the total protein, the membranes were stained with Ponceau S solution (0.1% (w/v) in 5% acetic acid (Sigma-Aldrich, St. Louis, MO, USA), after which membranes were rinsed prior to blocking in 5% BSA diluted in tris-buffered saline with 1% Tween (TBS-T) for 1 h at room temperature to block non-specific binding. Afterward, they were washed with TBS-T (3 × 5 min) and then incubated overnight at 4 °C with the appropriate primary antibody. Specific proteins were targeted using the following primary antibodies: total OXPHOS (1:500, #ab110413, Abcam, Cambridge, UK), PGC-1α (1:1000, #2178S, Cell Signaling Technology, Danvers, MA, USA), MitoBiogenesis cocktail (1: 250, #ab123545, Abcam), Mn-SOD (1:1000, #06-984, Merk, Darmstadt, Germany), NOX-2 (1:5000, #ab129068, Abcam), IRS-1 (1:1000, #3194, Cell Signaling Technology), p-IRS-1 (Ser789) (1:1000, #2389S, Cell Signaling Technology), AS160 (1:1000, #2670S, Cell Signaling Technology), p-AS160 (Thr642) (1:1000, #4288S, Cell Signaling Technology), GLUT-4 (1:1000, #ab33780, Abcam), PDK-4 (1:1000, #PA5-79800, Invitrogen), CD36 (1:1000, #ab133625, Abcam), CPTI (1:1000, sc-98834, Santa Cruz Biotechnology, Dallas, TX, USA), CaMKII (1:1000, #3362S, Cell Signaling Technology), p-CaMKII (Thr286) (1:1000, #sc-32289, Santa Cruz Biotechnology), Phospholamban (1:1000, #8495, Cell Signaling Technology), p-Phospholamban (Ser16/Thr17) (1:1000, #8496S, Cell Signaling Technology), SERCA2 ATPase (1:1000, #ab137020, Abcam, Cambridge, UK), Troponin I (1:1000, #4002, Cell Signaling Technology), p-Troponin I (Ser23/24) (1:1000, #4004S, Cell Signaling Technology), and GR-total (1:1000, #A303-491A, Bethyl Laboratories, Montgomery, TX, USA) as previously described [27,54,62,63,64,65,66,67]. All antibodies were prepared in 5% BSA in TBS-T and diluted to their final concentrations as per the manufacturer’s recommendations. The membranes were washed again with TBS-T (3 × 5 min) before being incubated for 1 h at room temperature with the appropriate Horse Radish peroxidase-labeled secondary IgG antibody (1:2000, anti-mouse #7076, anti-rabbit #7054, Cell Signaling Technology). Reactive proteins were detected using enhanced chemiluminescence with SuperSignal West Pico Chemiluminescent Substrate (Thermo Scientific, Waltham, MA, USA). The Western blot was captured using the ImageQuant LAS 4000 (GE Healthcare, Mascot, NSW, Australia), and protein abundance was quantified through densitometry using the ImageQuant TL 8.1 software (GE Healthcare, Mascot, NSW, Australia). The abundance of target proteins was adjusted relative to Ponceau S staining or specific reference proteins, vinculin (1:2000, #18799S, Cell Signaling Technology) or β-tubulin (1:1000, # 5346, Cell Signaling Technology).

### 2.6. Quantification of Cardiac Concentration of Glucocorticoid and Thyroid Hormones

Tissue hormone concentrations were measured using liquid chromatography (LC; Shimadzu Nexera XR, Shimadzu, Japan) combined with a SCIEX 6500 Triple-Quad system (MS/MS; SCIEX, US) following a modified protocol [35,68,69]. Initially, LV tissue (NM = 5M, 3F; ET = 5M, 5F; IVC = 5M, 6F; IVCHS = 4M, 5F; IVCHS+M = 5M, 4F) was homogenized in 500 μL 0.9% NaCl at 50 Hz for 2 min and then centrifuged at 12,000× *g* for 10 min at 4 °C. The supernatant (100 μL) was added to 300 μL acetonitrile containing the 50 ng/mL internal standard (cortisol-9,11,12,12-d4; Toronto Research Chemicals, Toronto, ON, Canada), vortexed for 1 min, and then centrifuged at 12,000× *g* for 10 min. The supernatant was transferred to a fresh Eppendorf tube, and the remaining pellet was resuspended in 300 μL of ethyl acetate. The mixture was vortexed for 1 min and then centrifuged at 12,000× *g* for 10 min. The supernatant was mixed with acetonitrile, gently inverted, and then evaporated to dryness using the GeneVac EZ-2 Evaporating System (GeneVac, Ipswich, UK). Dried samples were reconstituted in 50% methanol and then injected onto an ACQUITY UPLC BEH C18 Column (130 Å, 1.7 µm, 2.1 mm × 100 mm (Waters Corp, Milford, MA, USA)). Mobile phases were 0.1% formic acid in water (A) and 0.1% formic acid in acetonitrile (B). The flow rate was 0.3 mL/min, and mobile phase B was initially 10%, increased linearly to 90% over 10 min, and then held at 90% for 2 min, after which it returned to 10% for 3 min prior to injection of the next sample. Hormone concentrations were determined by integrating the data with a standard curve that ranged from 0.05 to 100 ng/mL. The conditions for detecting the analytes were as previously described [68].

### 2.7. Quantification of Cardiac Glycogen and Collagen Staining

Paraffin-embedded blocks of LV tissue, fixed with paraformaldehyde (NM = 5M, 4F; ET = 5M, 5F; IVC = 5M, 6F; IVCHS = 4M, 5F; IVCHS+M = 5M, 3F), were sectioned at 5 μm using a rotary microtome and mounted onto superfrost slides (VWR International, Radnor, PA, USA). Periodic acid–Schiff (PAS; a marker of glycogen content) and Masson’s trichrome (a marker of collagen content) staining was performed by University of Adelaide Histology Services. The slides were scanned at 40× magnification with a NanoZoomer-XR (Hamamatsu, Shizuoka, Japan) to capture images of the entire slides. PAS slides were analyzed with Fiji/ImageJ software (version 1.54f, NIH, Bethesda, MD, USA) using the color saturation threshold tool at 20× magnification, with five frames taken 1 mm apart. Masson’s trichrome slides were assessed using the Visiopharm software suite (version 2020.08, Denmark) with a custom threshold application at 10× magnification, covering the entire slide. Quantification of staining was validated through visual examination by a trained individual who was blinded to the treatment groups [69,70].

### 2.8. Statistical Analyses

Statistical analyses were conducted using GraphPad Prism 10 (GraphPad Software, Inc., Boston, MA, USA). Some samples were excluded from the analysis due to systematic or technical errors (hormone assay) or the absence of fixed tissue samples (histology). In the Western blot analysis, any non-quantifiable bands (due to an artifact on the band/s), were excluded and marked with an X on the blots. Up to one outlier per group was removed if identified using the Dixon Q test (α = 0.05; Minitab (v22.2.1)). The number of samples used for each analysis is provided in the table and figure legends. Data normality was tested using the Shapiro–Wilk test. The effects of treatment, sex, and their interaction were assessed with two-way ANOVA followed by Tukey’s multiple comparisons test. Data are presented as mean ± SD, and *p* < 0.05 was considered statistically significant.

## 3. Results

### 3.1. IVC Did Not Impact Cardiac Hormones and Glucocorticoid Receptors

The cardiac hormone concentration of cortisol, cortisone, cortisol:cortisone ratio, and corticosterone were not affected by treatment (Figure 1A–D). There were main sex effects in cortisol (*P_sex_* = 0.0006; Figure 1A) and cortisol:cortisone (*P_sex_* = 0.0022; Figure 1D) concentrations, such that females had higher concentrations compared to males. Testosterone concentration was affected by treatment (*P_trt_* = 0.0044) and sex (*P_sex_* < 0.0001), such that NM and IVCHS males had higher concentrations compared to males in the other groups (Figure 1E). Within each group, testosterone concentration was also higher in the males compared to the females (Figure 1E). The concentrations of T3, T4, and T3: T4 ratio were not affected by treatment or sex (Figure 1F–H). The protein expressions of GRα-A, GRα-C, and GRα-D3 were not affected by treatment or sex (Figure 1I–K).

### 3.2. IVC Reduced Mitochondrial Abundance and OXPHOS, Particularly in Males

The cardiac protein expression of OXPHOS complex 1 was affected by treatment (*P_trt_* = 0.0428) and sex (*P_sex_* = 0.0192), with males having higher expression compared to females and the expression in the NM group being higher than in the IVCHS group (*p* = 0.0439; Figure 2A). Complex 2 was not affected by treatment or sex (Figure 2B). Complex 3 expression was higher in males compared to females (*P_sex_* = 0.0194; Figure 2C). Complex 4 was affected by treatment (*P_trt_* = 0.0116; Figure 2D), with expression higher in the NM group compared to the ET group (*p* = 0.0396), IVCHS (*p* = 0.0235) and IVCHS+M (*p* = 0.0177) groups. Complex 5 was not affected by treatment or sex (Figure 2E). Mitochondrial abundance (MT-COXI:SDHA ratio) was higher in the males of the NM group than males of the ET (*p* = 0.0005), IVC (*p* = 0.0013), IVCHS (*p* = 0.0020), and IVCHS+M (*p* = 0.0039; Figure 2F) groups. Within the NM group, mitochondrial abundance was also higher in males compared to females (*p* = 0.0005; Figure 2F).

### 3.3. IVC Altered a Marker of Cardiac Metabolic Switch

The cardiac protein expression of p-IRS-1:IRS-1 was not affected by treatment or sex (Figure 3A). The expression of p-AS160:AS160 was lower in NM males compared to IVCHS males (*p* = 0.0214; Figure 3B). Cardiac expression of p-AS160:AS160 was also higher in IVC (*p* = 0.0414) and IVCHS (*p* = 0.0122) males compared to the females (Figure 3B). mRNA or protein expression of GLUT-4 and mRNA expression of *GSK3β* were not affected by treatment or sex (Figure 3C–E). Protein expression of PDK-4, a marker of cardiac metabolic switch, was affected by treatment (*P_trt_* = 0.0372), with expression higher in the NM group than in the IVCHS+M group (*p* = 0.0229; Figure 3F).

### 3.4. IVC Did Not Alter Markers of Cardiac Fatty Acid Metabolism

The cardiac mRNA and protein expression of CD36 was not affected by treatment or sex (Figure 4A,B). mRNA expression of *CPTI* was higher in females than males (*P_sex_* = 0.0060; Figure 4C), while CPTI protein expression was not affected by treatment or sex (Figure 4D). mRNA expression of *PPARα* was higher in NM females compared to IVCHS+M females (*p* = 0.0008) and higher in NM females compared to NM males (*p* = 0.0176; Figure 4E). mRNA expression of *PPARγ*, *ACC*, and *PC* was not affected by treatment or sex (Figure 4F–H).

### 3.5. IVC Reduced a Marker of Cardiac Antioxidant Defense, with No Effect on Markers of Contractility

The cardiac protein expression of p-CAMKII:CAMKII and p-PLN:PLN was not affected by treatment or sex (Figure 5A,B). The protein expression of SERCA2 was higher in NM males compared to NM females (*p* = 0.0105; Figure 5C). The protein expression of p-TroponinI:TroponinI was not affected by treatment or sex (Figure 5D). mRNA expression of *PGC-1α* was not affected by treatment or sex (Figure 5E), while its protein expression was higher in females compared to males (*P_sex_* = 0.0039; Figure 5F). Mn-SOD, a marker of cardiac antioxidant defense, was affected by treatment (*P_trt_* = 0.0173), such that the expression in the NM group was higher than in the IVCHS+M group (*p* = 0.0139; Figure 5G). Protein expression of NOX-2 was not affected by treatment or sex (Figure 5H).

### 3.6. IVC Did Not Alter Cardiac Glycogen and Collagen Content

The cardiac content of glycogen and collagen was not affected by treatment or sex (Figure 6K,L).

## 4. Discussion

This study revealed that IVC downregulated cardiac mitochondrial protein abundance in males, suggesting potential sex-specific disruptions to cardiac metabolism and function. Additionally, IVC reduced cardiac expression of OXPHOS complexes 1 and 4, which may indicate impaired energy production. Hearts from lambs subjected to IVCHS+M exhibited lower protein expression of PDK-4 and Mn-SOD compared to naturally mated controls, pointing to compromised energy efficiency and diminished defense to oxidative stress. These findings highlight the sex-specific and mechanistic impacts of two forms of IVC on postnatal cardiac health.

We found a sex-related difference in cardiac cortisol concentrations, with females showing higher concentrations compared to males, regardless of treatment. Data on sex-related differences in cardiac cortisol concentrations remain scarce; however, these differences may be influenced by sex hormones, such as estrogen. Research in human and various animal models have demonstrated that estrogen plays a key role in modulating the hypothalamic–pituitary–adrenal (HPA) axis [71,72]. In support of this interpretation, ovariectomy reduces basal cortisol, whereas estrogen replacement therapy increases free cortisol concentrations [73,74,75]. We also found that testosterone concentrations in the heart were higher in males compared to females, which may be associated with cortisol status. Androgens can modulate HPA axis activity [76], with evidence indicating that testosterone suppresses cortisol concentrations in men [77]. Furthermore, the lower testosterone concentrations observed in IVC males compared to the control males in this study suggest that IVC may also play a role in modulating this relationship.

GR isoforms are produced from a single GR gene (*NR3C1*) via alternative splicing and alternative translation initiation [78]. The human GR gene expresses two major mRNA variants: GRα and GRβ. The GRα mRNA further generates multiple isoforms including GRα-A, GRα-B, GRα-C1-3, and GRα-D1-3 [79,80]. Multiple GR protein isoforms have been detected in the placenta and lungs of both humans and sheep [27,28,29]; however, their expression and function in the heart are still largely unexplored. Recently, we have identified five GR isoforms (GRα-A, GR-P, GR-A, GRα-D2, and GRα-D3) in the fetal sheep heart, with their expression decreasing as gestational age advances [30]. Interestingly, the current study identified three isoforms (GRα-A, GRα-C, and GRα-D3) in adult hearts. This difference may reflect developmental regulation of GR isoforms, with certain isoforms being predominantly expressed during fetal development to meet the unique physiological and metabolic demands of the developing heart. Although information on the physiological functions and signaling pathways of each GR isoform is limited, studies have demonstrated that the translational isoforms of GRα (i.e., GRα-D3) exhibit comparable affinity for GCs and a similar ability to interact with glucocorticoid response elements (GREs) upon ligand activation [81,82]. The discovery that each GRα translational isoform regulates a distinct transcriptome suggests that cellular responses to GCs depend on the specific isoforms expressed [80]. For instance, while GRα-A and GRα-B isoforms are the most prevalent GR proteins in many cell types, trabecular meshwork cells in the human eye predominantly express the GRα-C and GRα-D isoforms [83]. In rodents, the GRα-C isoform is most abundant in the pancreas and colon, while the GRα-D isoform is most abundant in the spleen and lung [82].

This study revealed that IVC downregulated cardiac mitochondrial abundance, but only in males. Moreover, cardiac protein expression of OXPHOS complex 1 and 4 was lower in heart from IVC compared to naturally mated offspring. Mitochondria, often referred to as the “powerhouse” of the heart, play a vital role in producing ATP via OXPHOS [32,84]. Additionally, these organelles are involved in various signaling pathways associated with the regulation of oxidative stress, inflammation, mitophagy, calcium handling, and apoptosis: all key processes that contribute to cardiac dysfunction under various pathophysiological conditions [85,86,87]. Given the high energy requirements of the cardiac excitation–contraction and relaxation cycle, individuals with impaired OXPHOS are at a higher risk of developing heart diseases [88,89,90]. Recent studies have demonstrated that mitochondrial respiration is reduced in the hearts of fetal sheep with fetal growth restriction (FGR), suggesting that FGR increases the risk of developing heart disease and failure in adulthood [91]. Complex 1 deficiency is the most frequent mitochondrial disorder presenting in childhood and leads to cardiomyopathy later in life [92]. In line with our findings, Dimasi et al. also found that reduced in utero substrate supply leading to FGR was associated with decreased mitochondrial abundance and protein expression of OXPHOS complexes 2 and 4 in the fetal sheep heart [35], suggesting altered ATP production. Consistent with the findings of this study, they also found that cardiac mitochondrial abundance was higher in males compared to females, regardless of treatment [35]. Together, the observed decreases in mitochondrial abundance and OXPHOS complexes suggests that IVC may induce lasting mitochondrial alterations, with males showing greater susceptibility to these effects into adulthood.

We also found that cardiac protein expression of PDK-4 was lower in the IVCHS+M group compared to natural mate controls. PDK-4 functions as a metabolic switch, altering substrate utilization from glucose to fatty acids [93,94]. PDK-1 and PDK-4 are key heart isoenzymes [95] with distinct sensitivity to environmental stimuli and metabolic intermediates; PDK-1 primarily senses low oxygen, while PDK-4 responds to nutrient deprivation [96,97]. In end-stage systolic heart failure patients, protein expression of PDK-1 and PDK-2 remained unchanged in the LV, whereas PDK-4 expression was reduced by more than 60% [98]. Moreover, several studies have also reported metabolic shifts in heart failure, including reduced oxidative phosphorylation, decreased fatty acid oxidation, and increased reliance on glucose oxidation [99,100,101,102]. Taken together, the observed reduction in PDK-4 in the IVCHS+M hearts may suggest a metabolic shift favoring glucose oxidation over fatty acid oxidation. This may reflect a programmed disruption to metabolic flexibility, potentially impairing energy efficiency and cardiac performance, suggesting that methionine supplementation to mitigate the effects of IVCHS on heart growth may not be beneficial.

The cardiac protein expression of Mn-SOD was also lower in IVCHS+M hearts compared to NM controls in the current study. The antioxidant defense system is primarily composed of superoxide dismutase (SOD), catalase, glutathione peroxidases, and peroxiredoxins; deficiencies in these components can lead to oxidative stress [103]. In mammalian tissues, three isoforms of SOD are found in different locations: CuZn-SOD is located in the cytosol; Mn-SOD is found in the mitochondria; and extracellular SOD is present in the interstitial fluid [104]. Mitochondrial Mn-SOD is known as a cardioprotective enzyme that protects mitochondria and limits mitochondria-related apoptosis [105]. A lack of Mn-SOD in mice leads to dilated cardiomyopathy and triggers progressive heart failure due to excessive superoxide production and transcriptional changes in genes linked to heart failure [106]. Mitochondrial oxidative stress and damage may play a key role in the development and progression of LV remodeling and failure following myocardial infarction [107]. In the failing ventricular myocardium of end-stage heart failure patients, significant reductions in mitochondrial Mn-SOD protein abundance and activity have been observed [108]. Taken together, the reduced Mn-SOD abundance in the IVCHS+M hearts suggests impaired mitochondrial defense against oxidative stress, making them more susceptible to mitochondrial dysfunction and damage under physiological or pathological stress conditions. To support this, it has been indicated that during ART procedures, ovarian stimulation and *in vitro* embryo culture can increase oxidative stress, impair mitochondria, and reduce mitochondrial Mn-SOD abundance [109,110,111].

This study provides valuable insights into the long-term effects of IVC on cardiac metabolism and mitochondrial abundance in adolescent offspring; however, some limitations should be acknowledged. The IVC media used in this study, including human serum and methionine supplementation, are now considered outdated, as serum-free media have become the standard for preimplantation embryo culture across species [12,13]. Nonetheless, these findings remain relevant as they reflect IVC practices employed in the past, aligning with the context of human offspring now over 20 years of age. Additionally, the study did not assess mitochondrial OXPHOS activity, ROS production, proton leak, and mitochondrial structure, as these measurements require fresh tissue or specific fixation. Including these analyses in future studies will provide a more comprehensive evaluation of mitochondrial function. Future studies exploring mitochondrial bioenergetics (including ATP production), oxidative stress, and vascular analysis are also necessary to build upon these findings.

## 5. Conclusions

This study demonstrates that IVC, in the absence or presence of human serum with methionine supplementation, has a significant and sex-specific impact on cardiac metabolism in adolescent sheep. IVC downregulated mitochondrial abundance in male hearts and OXPHOS complexes in both sexes, potentially compromising energy production and cardiac function. Furthermore, reductions in PDK-4 and Mn-SOD protein abundance in the IVCHS+M hearts highlight disruptions in metabolic flexibility and antioxidant defenses, increasing susceptibility to oxidative stress and mitochondrial dysfunction. These findings underscore the lasting effects of IVC on cardiac health and the heightened vulnerability of males to these alterations, emphasizing the need for further research into IVC-induced programming of cardiometabolic health.

## Figures and Tables

**Figure 1 jdb-13-00017-f001:**
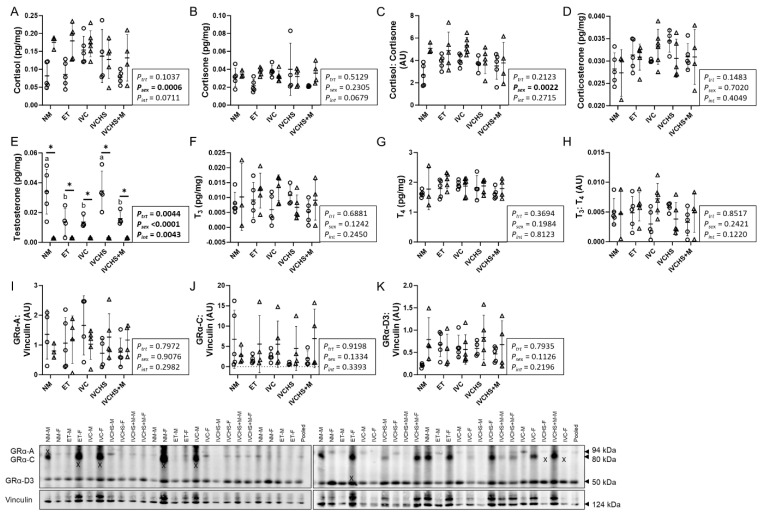
IVC did not impact cardiac hormones and glucocorticoid receptors. Males (M) = circles; females (F) = triangles. NM—natural mate (hormone = 5M, 3F; protein = 5M, 4F). ET—embryo transfer (hormone = 5M, 5F; protein = 5M, 5F). IVC—*in vitro* embryo culture (hormone = 5M, 6F; protein = 5M, 6F). IVCHS—*in vitro* embryo culture with human serum (hormone = 4M, 5F; protein = 4M, 5F). IVCHS+M—*in vitro* embryo culture with human serum and methionine supplementation (hormone = 5M, 4F; protein = 5M, 4F). (**A**) Cortisol, (**B**) Cortisone, (**C**) Cortisol: Cortisone, (**D**) Corticosterone, (**E**) Testosterone, (**F**) T_3_, (**G**) T_4_, (**H**) T_3_: T_4_, (**I**) GRα-A, (**J**) GRα-C, (**K**) GRα-D3. Hormone concentrations were measured using LC-MS/MS, and protein expression was assessed via Western blot. One outlier was excluded from each group, when applicable, using the Dixon Q test (α = 0.05). Data are presented as mean ± SD and analyzed using two-way ANOVA, followed by Tukey’s multiple comparisons test. Means with different superscript letters are significantly different (*p* < 0.05). (*) indicates mean values that are significantly different within the same group (*p* < 0.05). AU: arbitrary unit. (X) indicates data excluded from analysis (due to an artifact on the band/s).

**Figure 2 jdb-13-00017-f002:**
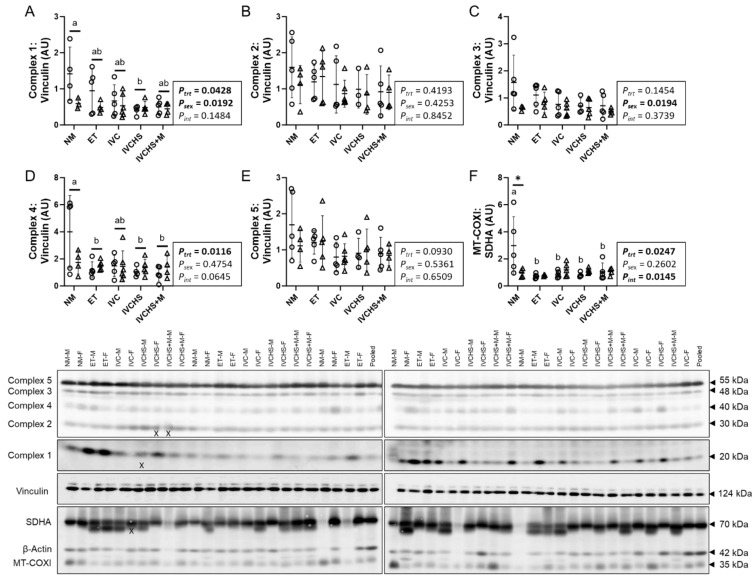
IVC reduced mitochondrial abundance and OXPHOS, particularly in males. Males (M) = circles; females (F) = triangles. NM—natural mate (protein = 5M, 4F). ET—embryo transfer (protein = 5M, 5F). IVC—*in vitro* embryo culture (protein = 5M, 6F). IVCHS—*in vitro* embryo culture with human serum (protein = 4M, 5F). IVCHS+M—*in vitro* embryo culture with human serum and methionine supplementation (protein = 5M, 4F). (**A**) Complex 1, (**B**) Complex 2, (**C**) Complex 3, (**D**) Complex 4, (**E**) Complex 5, (**F**) Mitochondrial content (MT-COXI: SDHA). Protein expression was assessed via Western blot. One outlier was excluded from each group, when applicable, using the Dixon Q test (α = 0.05). Data are presented as mean ± SD and analyzed using two-way ANOVA, followed by Tukey’s multiple comparisons test. Means with different superscript letters are significantly different (*p* < 0.05). (*) indicates mean values that are significantly different within the same group (*p* < 0.05). AU: arbitrary unit. (X) indicates data excluded from analysis (due to an artifact on the band/s).

**Figure 3 jdb-13-00017-f003:**
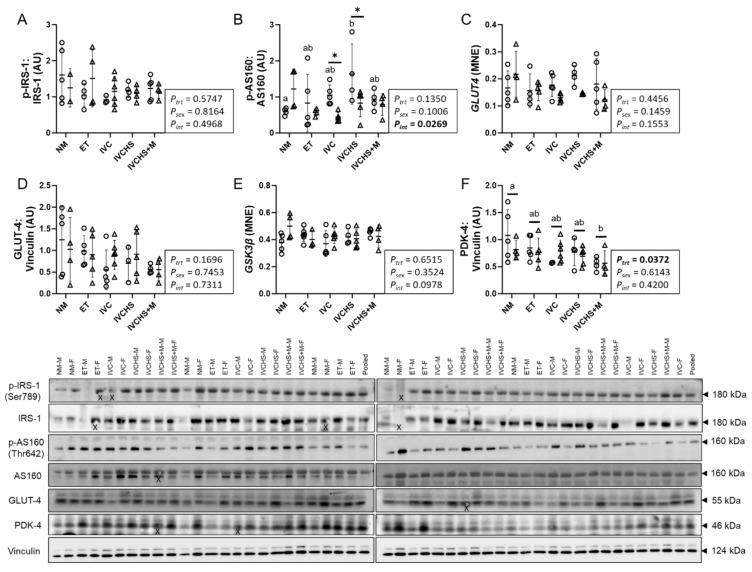
IVC altered a marker of cardiac metabolic switch. Males (M) = circles; females (F) = triangles. NM—natural mate (protein = 5M, 4F; mRNA = 5M, 4F). ET—embryo transfer (protein = 5M, 5F, mRNA = 5M, 5F). IVC—*in vitro* embryo culture (protein = 5M, 6F; mRNA = 5M, 6F). IVCHS—*in vitro* embryo culture with human serum (protein = 4M, 5F; mRNA = 4M, 5F). IVCHS+M—*in vitro* embryo culture with human serum and methionine supplementation (protein = 5M, 4F; mRNA = 5M, 4F). (**A**) p-IRS1: IRS-1, (**B**) p-AS160: AS160, (**C**) GLUT4 (gene), (**D**) GLUT4, (**E**) GSK3β (gene), (**F**) PDK-4. Protein expression was assessed via Western blot, and gene expression was analyzed using real-time RT-PCR. One outlier was excluded from each group, when applicable, using the Dixon Q test (α = 0.05). Data are presented as mean ± SD and analyzed using two-way ANOVA, followed by Tukey’s multiple comparisons test. Means with different superscript letters are significantly different (*p* < 0.05). (*) indicates mean values that are significantly different within the same group (*p* < 0.05). AU: arbitrary unit. (X) indicates data excluded from analysis (due to an artifact on the band/s).

**Figure 4 jdb-13-00017-f004:**
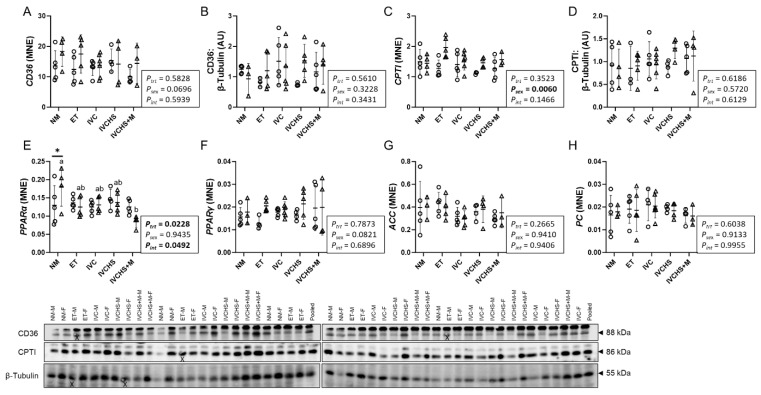
IVC did not alter markers of cardiac fatty acid metabolism. Males (M) = circles; females (F) = triangles. NM—natural mate (protein = 5M, 4F; mRNA = 5M, 4F). ET—embryo transfer (protein = 5M, 5F; mRNA = 5M, 5F). IVC—*in vitro* embryo culture (protein = 5M, 6F; mRNA = 5M, 6F). IVCHS—*in vitro* embryo culture with human serum (protein = 4M, 5F; mRNA = 4M, 5F). IVCHS+M—*in vitro* embryo culture with human serum and methionine supplementation (protein = 5M, 4F; mRNA = 5M, 4F). (**A**) CD36 (gene), (**B**) CD36, (**C**) CPTI (gene), (**D**) CPTI, (**E**) PPARα (gene), (**F**) PPARγ (gene), (**G**) ACC (gene), (**H**) PC (gene). Protein expression was assessed via Western blot, and gene expression was analyzed using real-time RT-PCR. One outlier was excluded from each group, when applicable, using the Dixon Q test (α = 0.05). Data are presented as mean ± SD and analyzed using two-way ANOVA, followed by Tukey’s multiple comparisons test. Means with different superscript letters are significantly different (*p* < 0.05). (*) indicates mean values that are significantly different within the same group (*p* < 0.05). AU: arbitrary unit. (X) indicates data excluded from analysis (due to an artifact on the band/s).

**Figure 5 jdb-13-00017-f005:**
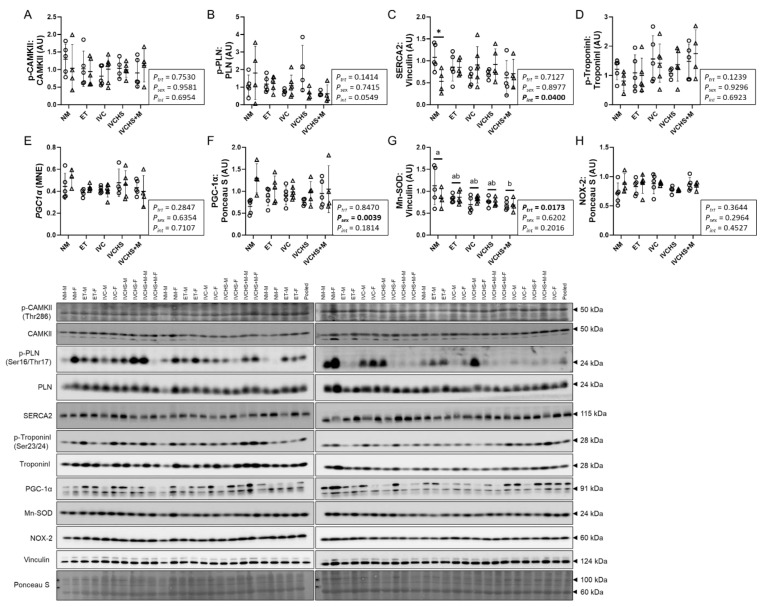
IVC reduced a marker of cardiac antioxidant defense, with no effect on contractility markers. Males (M) = circles; females (F) = triangles. NM—natural mate (protein = 5M, 4F; mRNA = 5M, 4F). ET—embryo transfer (protein = 5M, 5F; mRNA = 5M, 5F). IVC—*in vitro* embryo culture (protein = 5M, 6F; mRNA = 5M, 6F). IVCHS—*in vitro* embryo culture with human serum (protein = 4M, 5F; mRNA = 4M, 5F). IVCHS+M—*in vitro* embryo culture with human serum and methionine supplementation (protein = 5M, 4F; mRNA = 5M, 4F). (**A**) p-CAMKII: CAMKII, (**B**) p-PLN: PLN, (**C**) SERCA2, (**D**) p-TroponinI: TroponinI, (**E**) PGC1α (gene), (**F**) PGC1α, (**G**) Mn-SOD, (**H**) NOX-2. Protein expression was assessed via Western blot, and gene expression was analyzed using real-time RT-PCR. One outlier was excluded from each group, when applicable, using the Dixon Q test (α = 0.05). Data are presented as mean ± SD and analyzed using two-way ANOVA, followed by Tukey’s multiple comparisons test. Means with different superscript letters are significantly different (*p* < 0.05). (*) indicates mean values that are significantly different within the same group (*p* < 0.05). AU: arbitrary unit.

**Figure 6 jdb-13-00017-f006:**
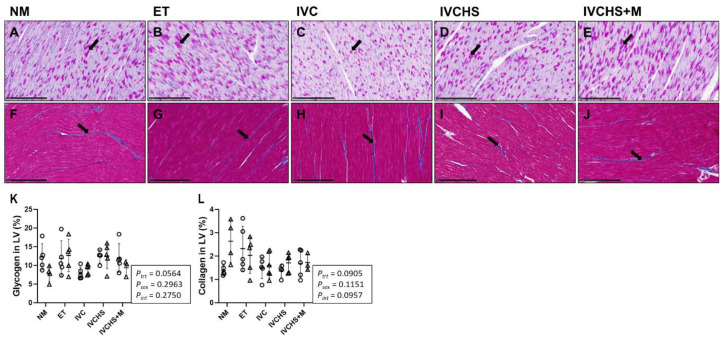
IVC did not alter cardiac glycogen and collagen staining. Representative micrograph (20× magnification) of glycogen staining using PAS (black arrow) in NM (**A**), ET (**B**), IVC (**C**), IVCHS (**D**), and IVCHS+M (**E**) groups. Representative micrograph (20× magnification) of collagen staining using Masson’s trichrome (black arrow) in NM (**F**), ET (**G**), IVC (**H**), IVCHS (**I**), IVCHS+M (**J**) groups. Glycogen (**K**) and collagen (**L**) content in the cardiac. Males (M) = circles; females (F) = triangles. NM—natural mate (histology = 5M, 4F). ET—embryo transfer (histology = 5M, 5F). IVC—*in vitro* embryo culture (histology = 5M, 6F). IVCHS—*in vitro* embryo culture with human serum (histology = 4M, 5F). IVCHS+M—*in vitro* embryo culture with human serum and methionine supplementation (histology = 5M, 3F). One outlier was excluded from each group, when applicable, using the Dixon Q test (α = 0.05). Data are presented as mean ± SD and analyzed using two-way ANOVA, followed by Tukey’s multiple comparisons test.

## Data Availability

All data supporting the results are presented in the manuscript.

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
