# Peer review of "In Vitro Embryo Culture Impacts Heart Mitochondria in Male Adolescent Sheep"

_jdb, 2025, doi:10.3390/jdb13020017_

Round 1

Reviewer 1 Report

Comments and Suggestions for Authors

1) Line 292-293: Please clarify whether Grubbs method was applied to whole treatment groups (NM, ET, IVC, etc.) or if it was individually applied to NM-male, NM female, ET male, ET female, etc categories, as represented in the graphs. If it's the latter, the test becomes highly unreliable as 'n' numbers are too low (less than at least n=6+).

2) Figure 2: Considering this is a key takeway from the study that informs the rest of the conclusions, it is important that further characterization is done on mitochondrial abundance, structure and activity. It would be good to see Electron Microscopy for mitochondria structure and abundance, and MitoTracker Red assay for mito activity, to solidify the conclusions.

3) Figure 5: Was there any increase in apoptosis or effect on proliferative activity in the IVC heart due to reduced mitochondrial abundance? As antioxidant defence is also reduced, it would be good to see TUNEL assays to measure impact on cardioprotectiveness.

4) Figure 6: Was there any impact on endothelial or capillary numbers? It would be nice to see PECAM or Lectin counts to round out the study.

Reviewer 2 Report

Comments and Suggestions for Authors

Dear authors,

Please see my comments below:

1. Please describe what type of assay was used for each figure either in the maintext or the figure legend.

2. Figure 2, was ATP level changed in cardiomyocytes? Should be a very easy assay if the authors still process the samples.

3. Figure 3, IVC altered marker of cardiac metabolic switch. Why is this important? Need to discuss. And again, did the altered metabolic switch affect ATP production?

4. Was cardiac function measured before euthanasia? e.g. ejection fraction. If was not clear if the difference in some of the marker affect the cardiac function.

Reviewer 3 Report

Comments and Suggestions for Authors

Manuscript ID: jdb-3499425
Title: In vitro embryo culture impacts heart mitochondria in male adolescent sheep
Authors: Reza Amanollahi, Stacey L. Holman, Ashley S. Meakin, Monalisa Padhee, Kimberley J. Botting-Lawford, Song Zhang, Severence M. MacLaughlin, David O. Kleemann, Simon K. Walker, Jennifer M. Kelly2, Skye R. Rudiger, I. Caroline McMillen, Michael D. Wiese, Mitchell C. Lock* and Janna L. Morrison*
Assisted reproductive technology has been widely used in human infertility treatment, livestock reproduction, laboratory animal maintenance and cryoresuscitation for biomedical research. The impact of gamete in vitro manipulation, especially in vitro culture, on the postnatal health has always been a concern. This manuscript described an original study on the effects of in vitro culture (IVC) of embryos on the postnatal health in adolescent sheep heart by the examination of several markers associated with cardiac metabolism, cardiac hormone, and GR receptors at the molecular level. The results demonstrated that the in vitro culture as well as the constitutes of the culture media impact postnatal heart health on males by effecting their cardiac health. This study provides more specific insights of the effects of IVC on postnatal health. The experiment design, execution and conclusion are sound. The manuscript is also well written. This manuscript can be accepted with minor revisions of several character spaces throughout the manuscript, i.e. 1: 1000 >> 1:1000 (L231, L235, L236, L252); 1min >> 1 min (L261) etc.

Round 2

Reviewer 1 Report

Comments and Suggestions for Authors

The statistical test used for outlier identification still raises concern. The Grubbs test may be more appropriate for much larger sample sizes (n>25, with n=6-8 being the minimum), and thus for n=4-5 samples might be inappropriate for applying this test. The Dixon test may be a more appropriate method for such small sample sizes. The authors are recommended to consult a statistician to ensure they are using the right method to exclude an outlier.

The authors have sufficiently addressed all other experimental concerns.
